# Session Rating of Perceived Exertion Is a Valid Method to Monitor Intensity of Exercise in Adults with Acute Burn Injuries

**DOI:** 10.3390/ebj6010004

**Published:** 2025-01-30

**Authors:** Joanne S. Page, Dale W. Edgar, Tiffany L. Grisbrook, Angela Jacques, Paul M. Gittings, Fiona M. Wood, Carly J. Brade

**Affiliations:** 1School of Allied Health, Faculty of Health Sciences, Curtin University, Bentley, WA 6102, Australia; joletterbox@yahoo.com.au (J.S.P.); tiffany.grisbrook@health.wa.gov.au (T.L.G.); angela.jacques@nd.edu.au (A.J.); carly.brade@curtin.edu.au (C.J.B.); 2State Adult Burn Unit, Fiona Stanley Hospital, Murdoch, WA 6150, Australia; paul.gittings@health.wa.gov.au (P.M.G.); fiona.wood@health.wa.gov.au (F.M.W.); 3Fiona Wood Foundation, Murdoch, WA 6150, Australia; 4Burn Injury Research Node, Institute for Health Research, The University of Notre Dame Australia, Fremantle, WA 6160, Australia; 5Burn Injury Research Unit, University of Western Australia, Crawley, WA 6009, Australia; 6UWA Medical School, Faculty of Health and Medical Sciences, The University of Western Australia, Perth, WA 6009, Australia; 7Perth Children’s Hospital, Perth, WA 6009, Australia; 8The Kids Research Institute Australia, Nedlands, WA 6009, Australia

**Keywords:** perception of exertion, RPE, training load, blood lactate, heart rate

## Abstract

Measuring exercise intensity for safety and to inform prescription in acute burn survivors, is challenging. This study aimed to assess the validity of adult patient end-of-workout rating of session perceived exertion (sRPE); and calculated training load (TL) (sRPE × session duration) as measures of exercise intensity. Secondly, the study aimed to compare clinician and patient perception of exercise effort during physiotherapist-led sessions. Repeated RPE data were collected every 5-min during two resistance exercise sessions completed by 25 burns patients. Physiological (heart rate [HR], blood lactate [BLa]) and perceptual measures (sRPE, ratings of pain, fatigue, delayed onset muscle soreness, sleep quality and stress) were also captured. Adjusted, multivariable linear regression models were used to determine the associations between sRPE and TL and significant predictor variables. Paired *t*-tests were performed to compare clinician and participant sRPE. Results: Average RPE calculated from 5-min repeats, after adjustment for age and %TBSA, was significantly associated with sRPE, *F*(1, 45) = 100.82, (*p* < 0.001, adjusted *R*^2^ = 0.64) and TL, *F*(1, 45) = 33.66, (*p* < 0.001, adjusted *R*^2^ = 0.39). No significant differences between patient and clinician sRPE were apparent (*p* = 0.948). Thus, one-off reporting of sRPE and calculated TL may be appropriate markers to monitor exercise intensity and aid prescription in individuals with burn injuries, regardless of patient and burn characteristics or time since burn. There was also no difference between patient and clinician’s perceptions of exercise effort.

## 1. Introduction

In an acute burn centre, there is limited capacity to hand control to patients within the environment. Exercise rehabilitation following burn injury is challenging and plays a vital role in the improvement and maintenance of physiological function, such as aerobic capacity and muscular strength [1]. A typical physiotherapy-led exercise session for individuals with burns may include the combination of aerobic, resistance, and balance exercises, in addition to stretching and body awareness activities [2]. In non-injured populations, it is important that the intensity of exercise is monitored to determine if the load is sufficient to stimulate the desired training adaptation without increasing the risk of injury, or overtraining [3,4,5]. In clinical populations, such as burn patients, monitoring intensity may further inform exercise prescription to ameliorate an albeit low risk of adverse events during or after exercise training, including any detriment to wound healing. Heart rate (HR) [6] and blood lactate (BLa) [7] are common objective physiological measures used to monitor exercise intensity. However, due to the hypermetabolic response experienced after burn, which is characterised by increased HR, respiratory rate and cardiac output, [8] it is unknown if these measures of exercise intensity are valid for an acute burns population.

Specifically, a study of 80 hospitalised burn patients, with >10% total body surface area (TBSA), found that 86.7% of the cohort had a resting BLa level measure of >2 mmol/L [9]. Further, increased serum lactate has been demonstrated after burn injury due to anaerobic oxidation of glucose associated with burn wound healing [10]. Heart rate can be significantly increased in children up to three years after burn compared to normal populations [11]. In adults, while acute heart rate elevation has been well established in burns > 10% TBSA [12], this has not been studied in smaller area burns or an acute exercise context. Consequently, altered physiological measures, HR and BLa, may be difficult to interpret in isolation to help prescribe exercise after burn injury. Thus, a clinically applicable, patient-centric method of monitoring exercise intensity is lacking and would be of benefit.

The Borg Rating of Perceived Exertion (RPE) scale is a frequently used patient reported measure of exercise intensity which rates momentary perceived exertion during exercise [13]. Session RPE (sRPE), reported via the category ratio 10 Borg scale (CR-10 RPE) [14], provides a rating of exertion for the exercise session as a whole [15]. Studies have shown sRPE to be a valid measure when tested against objective measures of exercise intensity such as percentage of HR reserve [15,16], methods of summated HR zone [17,18,19], percentage of peak rate of oxygen uptake (V̇O_2_) [15], percentage of HR peak [16] and average RPE [20]. Session RPE has been validated across different exercise modes including both aerobic [16,19] and resistance exercise in adults [21,22,23,24,25], overweight and obese children [26] and team sports [27]. Multiplying sRPE by duration of the training session in minutes quantifies the perceived training load (TL) for the session [15]. This calculation has been used to measure TL in non-injured sporting populations [19,28,29,30] and shown to be reliable in clinical populations such as chronic heart failure patients performing aerobic exercise [26,31]. It has also been demonstrated that TL (sRPE × session duration) is an effective method to quantify resistance training loads in adults with burn injuries, however it was not validated against physiological measures [3]. Additionally, it is unknown if sRPE or TL are suitable methods to monitor intensity of a physiotherapy-led exercise session after acute burn, particularly if the modes of exercise are mixed and varied.

Despite RPE being widely used, many extrinsic and intrinsic factors have the potential to influence it including pain, stress, fatigue, delayed onset muscle soreness (DOMS), sleep quality, environmental conditions, sex, age, fitness level, personality factors, and sociological factors [32,33]. However, these factors have not been studied during and in association with exercise in the acute burn population, where they may also be affected by the inflammatory response, and recovery from the injury. Individuals with burn injuries are likely to have increased levels of pain and anxiety which may affect their reported RPE and therefore sRPE [3]. Extrinsic and intrinsic factors also influence TL, as it is calculated from sRPE and duration of session. Grisbrook et al. [3] found that age, gender and %TBSA did not influence calculated TL with resistance training in individuals with burn injuries. However, pre-exercise pain significantly influenced TL, as increased pain was associated with higher sRPE. They noted that further research was required to determine the relationship between physiological measures of exercise intensity and sRPE in individuals with burn injuries [3].

Whilst the likelihood of overtraining and adverse events are low in a supervised, therapeutic exercise context, clinicians responsible for strength training should monitor exercise intensity to minimise risks for patients. There is a balance to be achieved with clinical exercise prescription to avoid any negative consequences and achieve training sufficient for positive adaption [4]. There is a potential to use sRPE and TL to assist clinicians to monitor and prescribe exercise load, and thus, it is important to understand whether a clinician and patient’s perception of exercise effort is similar. In a sporting context, coaches who underestimate players sRPE, leads to exercise training intensities which are too great and result in negative consequences and injury [34,35,36,37]. However, there have been no studies that compare clinician and patient perceptions of exercise intensity in a hospital setting.

Thus, this study aimed to assess if sRPE or TL have a relationship to objective measures of intensity (HR, within-session RPE and BLa) during a typical physio-led exercise session in adults with burn injuries. A secondary aim was to determine if %TBSA and other burn related characteristics, age, fatigue, DOMS, quality of sleep, levels of stress or pain were associated with the perception of effort during physiotherapy-led exercise sessions in adults with burns. Finally, this study aimed to determine if clinician and patient perception of exercise effort was similar.

## 2. Materials and Methods

### 2.1. Participants

Between February and March 2018, burn patients were invited to participate if they were aged 18 years or older and were receiving acute inpatient or post-discharge outpatient physiotherapy at the State Adult Burns Unit at Fiona Stanley Hospital (FSH). Participants were excluded if they had: poor English language skills, a current pregnancy, cognitive or neurological impairment, or any pre-existing musculoskeletal injuries preventing them from fully participating in exercise sessions. The study was approved by the South Metropolitan Health Service Human Research Ethics Committee (RGS503) and Curtin University Human Research Ethics Committee (HREC2017-0856). All participants provided written informed consent prior to participation.

### 2.2. Study Protocol

Data were collected from participants prior to and during two physiotherapy-led exercise sessions in the gym embedded in the FSH acute burn unit environment. These strength-focussed training sessions were individually prescribed by the treating physiotherapist and supervised by the physiotherapist and, or physiotherapy student. The sessions typically included after a warm-up, primarily resistance-based exercises interspersed with stretches, aerobic and balance activities, followed by a cool down.

#### 2.2.1. Patient Factors

To allow adjustment for known and potential confounders, participant demographic, injury and intervention information were recorded including: age, gender, %TBSA, days since burn injury, number of surgeries, days since last surgery and number of previous physiotherapy sessions. In addition, prior to the start of the first session, baseline anthropometric measures were assessed including height (Seca 0123 wall mount stadiometer, Seca, Hamburg, Germany) and body mass (A&D Medical UC-321PL Precision Health scale, A&D Medical, Ann Arbor, MI, USA).

#### 2.2.2. Perceived Exertion Measures

Prior to starting the first session participants were introduced to the RPE category-ratio scale (CR10-RPE); a self-reported scale ranging from 0–10 where zero represents no exertion (nothing at all) and 10 represents maximal exertion (extremely strong, maximal) [14]. Participants were informed when a rating was to be obtained and they silently provided their rating to the researcher during the session via a hard copy, printed scale. Due to expected increase in variability after burn injury, ratings of perceived exertion were measured before, approximately every five minutes during, and immediately after the exercise session using the CR10-RPE. The participant indicated their real-time RPE, non-verbally to the tester by pointing to a printed CR10-RPE scale to ensure the treating clinician remained blinded to the response. The exertion ratings were captured between sets, irrespective if the prescribed set of exercise repetitions had been completed. Average RPE was then calculated to ‘summarise’ exertion for all exercises completed during the whole session as a key comparator for the one-off patient-perceived exertion for each session.

Ten minutes after the completion of the session, participants were shown the CR10-RPE scale and asked to provide a rating of the overall difficulty of the entire session (sRPE), by answering “How hard do you feel you were working during the training session as a whole?”. The participant’s sRPE was then multiplied by the duration of the exercise session (minutes) to ascertain a TL for the session. This information was given confidentially to the tester so as to not influence the clinician’s perception of the participant’s sRPE.

In addition, each patient’s well-being was monitored using Hooper’s Index. The index is derived as a numerical quantification of self-analysis questionnaires involving subjective ratings of four factors: fatigue, levels of stress, delayed onset muscle soreness (DOMS) and sleep quality/disorders [32], was collected prior to the start of each exercise session. This method uses a scale of 1–7 (very low or good indicated by 1 to very high or bad indicated by 7) with the summation of the four ratings recorded as a composite indicator of these factors [32]. The participant’s level of pain was recorded prior to the exercise session using a visual analogue scale (VAS) [38]. The pain VAS is a unidimensional numerical rating scale of pain intensity constructed along a straight horizontal line of fixed length, with two endpoints whereby zero (0), denoted “no pain at all” and 10, described “worst imaginable pain” [38]. At completion of the session, the highest level of pain during the session was also recorded.

#### 2.2.3. Physiological Measures

To monitor HR, participants were fitted with and wore a Polar FT4 monitor HR (Polar Electro Oy, Kempele, Finland) for the duration of the exercise session. Heart rate was recorded before, every five minutes during, and immediately after the exercise session. Average HR (ӿHR), peak HR (pHR), and peak HR as a percentage of age predicted HR max (pHR_%age)_ (peak HR/(208 − 0.7 × age) × 100) were calculated [39] for the whole session.

Blood lactate was measured from a 0.3 µL capillary whole blood sample attained from the finger (ARKRAY Lactate ProTM 2, ARKRAY, Kyoto, Japan) before (resting) and immediately after the session (post exercise). The sample was taken from the ear lobe if unable to attain a blood sample from the finger, or where the participants’ hands were injured by a burn. The change in BLa (post exercise BLa − resting BLa), and percentage of BLa change from baseline [((post exercise BLa − resting BLa)/resting BLa) × 100] was calculated for the whole session.

#### 2.2.4. Perceived Exertion Comparison

After completion of the exercise session, the supervising clinician was asked to provide a single rating of their perception of the participant’s exertion for the whole session to the researcher, independent to the patient rating and recorded confidentially so as to not influence the patient’s reported sRPE. The clinician’s level (qualified physiotherapist or physiotherapy student) and years of experience were also recorded.

### 2.3. Data Analysis

A sample of 24 participants was estimated a priori using published session RPE dispersion values [16], to provide 80% power (two-tailed) to detect a beta coefficient (β) of 0.51 in a linear regression model or a moderately large correlation coefficient (rho = 0.5). (G*Power 3.1.9.7). Descriptive statistics included frequency distributions for categorical variables and means, standard deviations, medians and interquartile ranges, according to normality, for continuous variables. Normality testing included assessment of histograms and Shapiro-Wilk tests. The validity of the outcome variables sRPE and TL was assessed by exploring criteria-based associations with known predictor variables which included average HR for session, peak HR, peak HR as a percentage of age predicted HR max, average RPE, the change in BLa, post exercise BLa, and percentage of BLa change from baseline. The influence of independent covariates was also examined and included participant characteristics and clinical measures such as age, %TBSA, gender, height, weight, body mass index (BMI), days since burn injury, number of surgeries, days since last surgery, number of physiotherapy-led exercise sessions, fatigue, stress, DOMS, sleep quality, Hooper’s Index, pre-exercise pain and highest pain during the session.

Spearman’s Rho correlation coefficients were used to examine correlations between raw outcome variables, predictor variables and independent covariates. Significantly correlated variables were analysed using univariate linear regression analysis. Independent covariates were regressed against both predictor and outcome variables in order to identify confounding variables. Independent covariates that demonstrated statistically significant association with both outcome and predictor variables were considered as confounders for inclusion in multivariable models.

Multivariable linear regression models were then used to determine the associations between sRPE and TL and significant predictor variables, adjusting for confounders (as specified, and other clinically relevant independent covariates. Individual models were run for sRPE and TL. The final multivariable models included only variables shown to significantly and independently influence TL and sRPE. Results of the linear regression analysis were reported as regression coefficients (β), standard error (SE) and *R*^2^.

Paired *t*-tests were performed to compare clinician and patient sRPE, overall and stratified into two categories: physiotherapy student and qualified physiotherapist.

All statistical analyses were undertaken using IBM SPSS Statistics for Windows, Version 25.0. (Armonk, NY, USA: IBM Corp). All hypothesis tests were two-sided and *p*-values less than 0.05 were considered statistically significant for all analyses.

## 3. Results

### 3.1. Participants

Twenty-five participants with burn injuries (21 males, 4 females) with mean age 45 ± SD = 17.4 years (range: 19–81 years); mean %TBSA of 6 (range: 0.25–27%) were recruited into the study. Participant characteristics and physiological and perceptual measures are outlined in Table 1 and Table 2, respectively.

Twenty-four (24) participants completed two physiotherapy-led exercise sessions, and thus, data were included from 49 sessions in total. The mean number of repeated RPE measurements taken during Session 1 and 2 was 4.1 (range 0–8) and 4.3 (range 0–10), respectively, excluding the pre and post-session collections.

Thirteen clinicians (10 females, three males) including 11 qualified physiotherapists and two physiotherapy students supervised the physiotherapy-led exercise sessions. The qualified physiotherapists had a mean of 12.6 years’ experience (range: 2.5–25 years).

### 3.2. Univariate Analysis

Outcome, predictor, and independent variables were analysed for correlations (Table 3). No common independent covariates were identified between the outcome and predictor variables.

Average RPE was the only predictor shown to have a statistically significant association with sRPE, *p* <0.001 (Appendix A). In contrast, predictors: peak HR, peak HR as a percentage of age predicted HR max, average RPE, and post exercise BLa were all significantly associated with TL (Appendix A).

### 3.3. Multivariable Analysis

After adjusting for age and %TBSA, average RPE remained a significant predictor of sRPE, *F*(1, 45) = 100.82, *p* < 0.001, *R*^2^ = 0.64 (Table 4). For every (1) level increase in average RPE, there was a 1.14 point increase in sRPE.

In modelling TL, average RPE and post exercise BLa were initially included but after adjusting for age and %TBSA [2,35], the latter was excluded from the final model. After adjustments, the final regression model showed that average RPE, was a significant predictor of TL, *F*(1, 45) = 33.66, *p* < 0.001, adjusted *R*^2^ = 0.39 (Table 5). For every (1) level increase in average RPE, there was a 38.78 point increase in TL (*p* < 0.001).

### 3.4. Clinician Compared with Patient sRPE

There was no significant difference between patient sRPE (4.60 ± 1.76) and clinician sRPE (4.92 ± 1.66) (*p* = 0.948). After stratifying by physiotherapy students or qualified physiotherapists there remained no significant difference to the patients’ sRPE (*p* = 1.000 and *p* = 0.903, respectively).

## 4. Discussion

The results of the present study demonstrated that a single patient or clinician reported session RPE and calculated training load (sRPE × duration) were valid measures of exercise intensity in burn patients with up to ~25% TBSA. This clinically applicable finding is primarily related to the statistically and clinically relevant association of average RPE, arising from repeated patient within-session ratings of exertion during exercise rehabilitation workouts. This is the first study to investigate the association between sRPE and physiological and perceptual variables measured during customised physiotherapy-led strength training session in individuals with a burn injury. Point-of-care and real-time change of physiological measures HR and BLa showed no evidence of associations with sRPE or TL during this study. Further sRPE and TL were not associated with any patient or burn characteristics in this study, which strengthens clinical applicability of our results. Lastly, the study found that there was no evidence of differences between patients’ actual and the supervising physiotherapist’s estimation of patient sRPE, again enhancing clinical applicability of the study findings. This finding also highlights the importance of developing trust and rapport in therapeutic relationships and a key strategy to help motivate acute burn patients when completing challenging exercise rehabilitation sessions.

The multivariable analysis demonstrated that average RPE was the only predictor variable associated with sRPE. After adjusting for age and %TBSA, average RPE accounted for 64% of sRPE variability and the correlation between sRPE and average RPE was high (rho = 0.82, Table 3). This is in line with Foster et al. [20] who compared sRPE with average RPE (measured every 10 min) during a 60 min exercise session in 12 well-trained adults, where average RPE accounted for 79% of sRPE. Apart from sessions where the average RPE was very high (>7), they found a good correspondence between average and sRPE [20]. They concluded that sRPE was a valid surrogate for the average RPE, therefore markedly reducing the patient report burden which is also positive for the acute burn patient context. McGuigan et al. [26] also demonstrated a high correlation (r = 0.88) between sRPE and average RPE in overweight and obese children performing resistance exercise. Despite these findings of high correlation between sRPE and average RPE it has consistently been reported that sRPE is significantly higher than average RPE for a given exercise bout in a variety of populations [20,26,40,41,42]. The current study also demonstrated that mean sRPE was higher than mean average RPE (Table 2). It has been suggested that the final RPE taken within the exercise session (terminal RPE), may influence sRPE and bias sRPE upward [20,41]. However, Hornsby et al. [41] and Green et al. [40] demonstrated that terminal RPE was not linked with sRPE in recreationally active adults. It has also been proposed that increases in sRPE are associated with increased exercise duration; however, this was refuted by Green et al. [40] as they demonstrated that the association of session duration on sRPE was minimal and not significant in healthy adults, with exercise intensity being a stronger mediator of sRPE [40]. The current study is the first to conduct a validation analysis in a clinical population where additional rest periods may be required such that session duration may not truly reflect increased workload. It is also possible that accumulated fatigue or tiredness may have contributed to the increased sRPE [26,42]. The results of this study suggest that these factors require further investigation as while the average RPE was significantly associated with TL, it only accounted for 39% of TL. The correspondence between average RPE and TL is likely to be lower due to the influence of session duration on the TL calculation, which is not considered in the calculation of average RPE. Nevertheless, the current results agree with previous findings that while sRPE does not directly reflect average RPE responses, it is a valid and rapid method that can be used to monitor global intensity and volume of a physical exercise session [41].

In the current study sample, neither HR nor BLa remained significantly associated with sRPE or TL after adjusting for burn and host metabolic factors, age and %TBSA. Although peak HR as a percentage of age predicted HR max had significant individual influence on TL in the univariate analysis, comparisons are difficult as usually TL is compared to other TL measures for example the Banister’s TRIMP [4]. Blood lactate in the final model had no association with either TL or sRPE however post BLa was significant in the univariate analysis for TL (Appendix A). In non-injured exercisers, it is well established that relationships between HR (for example, % HR reserve, summated HR zones and %HR peak) and sRPE in steady state and non-steady state exercise exist [15,16,19]. Due to various factors which alter HR after acute burn, contrasting results are not unexpected due to different measures and assumptions for HR comparisons (e.g., peak HR vs %HR reserve) and methods of calculation (%HR peak derived from maximal oxygen consumption [16] compared to %HR peak from estimated age predicted HR max). Green et al. [43] found that for high intensity interval cycling, HR and RPE had better correspondence than RPE and BLa due to differences in response times in physiological variables. Similarly, this could account for the lack of association with HR and BLa and sRPE and TL in the current study in addition to constantly changing exercise modalities (resistance, steady state aerobic and low intensity activities like stretching). Also, lower mean patient peak HR (110 vs. 164 bpm) and post BLa (6.6 vs. 5.5 mmol/L) were recorded in the current study compared to Green et al. [43], respectively. The real-world clinical nature of the study in acute burn patients likely contributed to the difficulty of confirming strong associations as each physiotherapy-led exercise session was designed based on the patient’s daily disposition and current impairments, such that no two sessions were the same.

Participant characteristics examined in this study (age, gender, height, weight, BMI, %TBSA, days since burn injury, number of surgeries, days since last surgery, and number of previous physiotherapy sessions) did not correlate with sRPE or TL. This finding is in agreement with Grisbrook et al.’s [3] study years after acute discharge which found that participant characteristics such as age, gender and %TBSA were not associated with TL during resistance training in individuals with burn injuries. Despite differing exercise protocols used in the respective studies these findings suggest that sRPE is a valid method of quantifying intensity during a physiotherapy exercise session which includes resistance exercise. Furthermore, no association between sRPE and TL and measures of well-being (Hooper’s Index) were found in the present study which is somewhat in agreement with Haddad et al. [32] who found the same measures were not significantly associated with RPE following 10 minutes of submaximal exercise in junior soccer players. Despite the differing durations and modalities of exercise, these studies demonstrated that well-being measures in different populations (healthy and clinical) may not influence sRPE and RPE. As these factors (fatigue, stress, DOMS, poor sleep) are seemingly commonplace in those with a burn injury, there was no evidence to support their influence on exercise effort, therefore these may be less important considerations when prescribing exercise daily in this population.

In contrast, the current study did not show an association between baseline pain or peak pain and sRPE or TL, whereas a study by Grisbrook et al. [3] found that pre-exercise pain significantly influenced TL (as calculated by sRPE × duration). Despite comparable pain ratings (current study = 2.24; Grisbrook et al. exercise group = 2.05) between the studies the differing exercise modalities and interaction with the time since burn may be implicated in these opposing findings such that the resistance training in the Grisbrook et al. [3] study may have been of a higher intensity compared to the current study where the TL was based on the warm-up, aerobic and resistance components as well as the stretching and cool-down. The timing of the exercise session may have also influenced the results. In the current study, the exercise sessions were conducted on average 28 days (range = 1–284 days) post burn injury, whereas in Grisbrook et al.’s [3] study the first exercise session occurred within 72 h of injury, which may have influenced the patient’s perception of effort. If increased pre-exercise pain contributes to a higher perception of exercise effort (as found by Grisbrook et al. [3]), this needs to be considered during exercise prescription, so that exercise duration is altered accordingly, to ensure overtraining does not occur. Further research needs to evaluate when pre-exercise pain, which can be apparent in a population with burns, needs to be considered in exercise prescription.

The agreement between the clinicians and patients sRPE is important. To ensure the clinicians’ intended or prescribed exercise intensity is the same as what is being perceived by the patient. While caution is warranted with the paired sample (n = 49), there was no evidence of significant difference between clinicians and participants sRPE. Whilst this is the first study to compare clinician and patient perception of effort, several studies have compared coaches’ and athletes’ sRPE. Brink et al. compared the sRPE of coaches and elite soccer players [34]. They found that players consistently perceived training sessions as harder than what was intended by the coach [34] which is supported in other sports [35,36,37,44]. In the current study all sessions were conducted one on one, which may have contributed to the similarity in clinician and participant sRPE post-session ratings. A planned rating of the session is difficult in an exercise rehabilitation environment such as after acute burn, as the session may need to be altered due to patient compliance with and ability to complete pre-planned activities. Thus, further investigation is warranted to explore if a clinician’s intended training exertion (before the session) matches a patient’s perceived exertion (after the session), to enhance the use of sRPE as a method of exercise prescription in patients with burn injuries, especially in the context where patients can self-monitor exercise intensity after discharge from the Burns service using their sRPE.

## 5. Conclusions

Session RPE and training load are inexpensive and simple to use methods of monitoring exercise intensity, and this study showed they have valuable clinical utility in day-to-day exercise prescription and practice after acute burn injury. Similar clinician and patient sRPE reports in this study increase the confidence that patient sRPE can inform exercise prescription for individuals and potentially increase their control while recovering from burn injuries in a hospital setting.

## Figures and Tables

**Table 1 ebj-06-00004-t001:** Burn injured participant characteristics.

Characteristic	Session 1 (n = 25) Median (IQR)	Session 2 (n = 24) Median (IQR)	Total Cohort Median (IQR) (Range)
Age (y)	51 (22.5)	52 (30)	52 (34) (19–81)
Body Weight (kg)	87.7 (29.5)	85.95 (28.5)	87.6 (28.5) (63.2–145.6)
Height (m)	1.69 (0.13)	1.69 (0.15)	1.69 (0.15) (1.55–1.82)
BMI	30.67 (6.09)	30.07 (5.74)	30.2 (5.69) (24.69–53.48)
TBSA (%)	3 (5.8)	3 (7)	3.25 (7.32) (0.25–26.95)
Days since burn	5 (7)	10 (11)	9 (10) (1–284)
Number of surgeries	0 (1)	1 (0)	1 (1) (0–4)
Days since last surgery	0 (7)	4 (5)	2 (7) (0–102)
Number of PT sessions	1 (7)	3 (7)	2 (7) (0–76)
Resting HR	88 (19)	78 (21)	87 (46) (58–141)
Resting BLa	2 (1.9)	2.1 (1.82)	2.1 (1.7) (0.8–9)

Abbreviations: BMI = body mass index, TBSA = total body surface area, PT = physiotherapy, HR = heart rate, BLa = blood lactate.

**Table 2 ebj-06-00004-t002:** Physiological and perceptual measures taken from the participants with burn injury during their physiotherapy session, presented as means (SD).

	Session 1 (n = 25)	Session 2 (n = 24)
Average Session Duration (mins)	34.08 (10.3)	33.92 (11.66)
Resting HR	92.36 (14.27)	84.96 (21.40)
Average HR	116.5 (19.34)	103.55 (21.59)
Peak HR	128.44 (24.42)	117.38 (24.33)
Peak HR as a % of age predicted HR max	72.90 (13.66)	64.14 (18.93)
Baseline Pain	2.24 (2.28)	3.13 (2.69)
Peak Pain	5.22 (2.36)	5.92 (2.55)
Hoopers Index	12.88 (3.91)	12.58 (4.18)
Resting BLa (mmol/L)	2.75 (1.92)	2.37 (1.45)
Post exercise BLa (mmol/L)	5.85 (3.98)	5.24 (3.48)
Change in BLa (mmol/L)	3.10 (4.19)	2.88 (3.31)
BLa percentage change from baseline	188.45 (248.28)	171.18 (199.21)
Session duration (mins)	34.08 (10.30)	33.92 (11.66)
Average RPE	4.10 (1.28)	4.28 (1.61)
sRPE	4.60 (1.76)	5.08 (2.17)
Training load	153.96 (71.89)	170.96 (98.52)

Abbreviations: HR = heart rate (beats per minute), BLa = blood lactate, RPE = ratings of perceived exertion.

**Table 3 ebj-06-00004-t003:** Correlations (Spearman’s Rho) between participant characteristics and predictor and outcome variables.

	Training Load	sRPE	HR Average for Session	Peak HR	Peak HR as a Percentage of Age Predicted HR Max	RPE Average for Session	BLa Change (mmol/L)	Post Exercise BLa (mmol/L)	BLa Percentage Change from Baseline
Age (years)	0.134	0.167	−0.015	−0.072	−0.222	0.163	**0.291 ***	0.155	**0.402 ****
Body Mass (kg)	0.219	0.274	−0.129	−0.208	0.053	0.28	0.02	0.060	0.07
Height (m)	0.115	0.187	−0.137	−0.161	0.105	0.016	0.148	0.066	0.161
BMI	0.197	0.265	−0.014	−0.125	−0.092	**0.355 ***	0.021	0.041	0.091
TBSA (%)	**0.284 ***	0.204	**0.452 ****	**0.461 ****	**−0.413 ****	0.193	−0.007	0.113	−0.185
Days since burn	**0.326 ***	0.179	−0.078	0.005	0.041	0.234	0.131	0.115	0.057
Number of Surgeries	0.259	0.2	0.099	0.127	−0.118	**0.282 ***	0.192	0.129	0.128
Days since last surgery	0.268	0.099	0.063	0.128	−0.112	0.167	0.217	0.118	0.186
Number of PT sessions	**0.299 ***	0.133	0.143	0.165	−0.191	0.25	0.221	0.146	0.149
Fatigue	−0.134	0.013	0.12	−0.003	−0.013	0.141	−0.018	0.014	−0.005
Stress	0.19	0.212	0.066	−0.001	0.011	**0.394 ****	−0.202	−0.105	−0.188
DOMS	−0.128	0.070	**−0.290 ***	**−0.347 ***	**0.387 ****	0.085	**−0.396 ****	**−0.339 ***	**−0.310 ***
Sleep Quality	0.053	−0.057	−0.105	−0.132	0.178	0.169	−0.176	−0.09	**−0.292 ***
Hooper’s Indices	−0.014	0.1	−0.065	−0.172	0.205	0.277	**−0.310 ***	−0.223	**−0.301 ***
Training Load		**0.788 ****	0.135	0.258	−0.279	**0.666 ****	0.101	0.254	0.018
sRPE	**0.788 ****		0.085	0.194	−0.229	**0.822 ****	0.072	0.25	−0.031
RPE Peak	**0.748 ****	**0.769 ****	0.212	0.252	−0.281	**0.888 ****	0.059	0.223	−0.022
RPE average for session	**0.666 ****	**0.822 ****	0.21	0.237	**−0.285 ***		0.11	0.275	−0.004
HR average for session	0.135	0.085		**0.905 ****	**−0.888 ****	0.21	0.258	**0.297 ***	0.106
Peak HR	0.258	0.194	**0.905 ****		**−0.943 ****	0.237	**0.380 ****	**0.482 ****	0.193
Peak HR as a percentage of age predicted HR max	−0.279	−0.229	**−0.888 ****	**−0.943 ****		**−0.285 ***	**−0.485 ****	**−0.548 ****	**−0.340 ***
Pre exercise BLa (mmol/L)	0.254	**0.296 ***	0.171	0.219	−0.115	**0.349 ***	**−0.285 ***	0.24	**−0.599 ****
BLa Change (mmol/L)	0.101	0.072	0.258	**0.380 ****	**−0.485 ****	0.11		**0.776 ****	**0.897 ****
Post exercise BLa (mmol/L)	0.254	0.25	**0.297 ***	**0.482 ****	**−0.548 ****	0.275	**0.776 ****		**0.593 ****
BLa percentage change from baseline	0.018	−0.031	0.106	0.193	**−0.340 ***	−0.004	**0.897 ****	**0.593 ****	
Resting Pain	−0.212	0.080	0.01	−0.021	0.089	0.091	−0.212	−0.105	−0.226
Highest Pain	0.093	0.224	−0.275	−0.256	0.237	0.236	−0.099	−0.127	−0.035

Abbreviations: BMI = body mass index, TBSA = total body surface area, PT = physiotherapy, DOMS = delayed onset muscle soreness, HR = heart rate (beats per minute), RPE = ratings of perceived exertion, BLa = blood lactate. *p* < 0.05 *, *p* < 0.01 **.

**Table 4 ebj-06-00004-t004:** Final multiple regression model for session RPE.

Independent Variable	*β*	*SE_β_*	*p* Value
Average RPE *	1.136	0.113	**<0.001**
Age	0.006	0.009	0.511
TBSA (%)	−0.005	0.023	0.832

Abbreviations: RPE = ratings of perceived exertion, TBSA = total body surface area. *β* = unstandardized regression coefficient: *SE_β_* = standard error of the coefficient. * Average RPE calculated from all 5-min within-session patient reported RPE.

**Table 5 ebj-06-00004-t005:** Final multiple regression model for Training Load (TL = session RPE × session duration).

Independent Variable	*β*	*SE_β_*	*p* Value
Average RPE	38.779	6.684	**<0.001**
Age	0.074	0.550	0.894
TBSA (%)	1.444	1.340	0.287

Abbreviations: RPE = ratings of perceived exertion, TBSA = total body surface area. *β* = unstandardized regression coefficient: *SE_β_* = standard error of the coefficient.

## Data Availability

De-identified data will only be available upon request to the corresponding author, and with appropriate Ethics approvals.

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
