# Peer review of "Session Rating of Perceived Exertion Is a Valid Method to Monitor Intensity of Exercise in Adults with Acute Burn Injuries"

_2673-1991, 2025, doi:10.3390/ebj6010004_

Round 1

Reviewer 1 Report

Comments and Suggestions for Authors

The authors have conducted a thorough investigation into the use of session ratings of perceived exertion (sRPE) and training load (TL) as measures of exercise intensity in burn patients. The methodology is robust, and the statistical analysis is sound, providing valuable insights into the applicability of sRPE and TL in a clinical setting. The findings that sRPE and TL are valid measures for monitoring exercise intensity in this population are particularly noteworthy and contribute meaningfully to the field of burn rehabilitation.

While the study is well-executed, I have a few questions that I believe should be addressed to further strengthen the manuscript:

  1. Line 46: “In clinical populations such as burn patients, monitoring intensity may also help to inform exercise prescription and reduce the risk of adverse events and impeded wound healing.”

    • This sentence seems to imply that there are frequent adverse events and that wound healing can be impeded by exercises, which is not often the case. You come back to this in line 97. Could you change this sentence?
  2. Line 184:

    • Which outcome parameter was used to calculate power (the mean RPE or the TL)? Was this calculated in advance?
  3. Line 66 and in the Methodology Section:

    • Were there any specific protocols or training provided to the participants to standardize their responses to the RPE?
  4. Line 365-380:

    • It is indeed curious that no association was found between pain and TL in your study. Exercise can also have a pain-reducing effect; could this have contributed to your findings?

This study provides valuable insights that can improve the monitoring and prescription of exercise in burn rehabilitation, ultimately benefiting patient outcomes. 

Author Response

Reviewer #1

The authors have conducted a thorough investigation into the use of session ratings of perceived exertion (sRPE) and training load (TL) as measures of exercise intensity in burn patients. The methodology is robust, and the statistical analysis is sound, providing valuable insights into the applicability of sRPE and TL in a clinical setting. The findings that sRPE and TL are valid measures for monitoring exercise intensity in this population are particularly noteworthy and contribute meaningfully to the field of burn rehabilitation.

DE – Thank you to the reviewer for these comments. These and the positive feedback below are much appreciated.

While the study is well-executed, I have a few questions that I believe should be addressed to further strengthen the manuscript:

  1. Line 46: “In clinical populations such as burn patients, monitoring intensity may also help to inform exercise prescription and reduce the risk of adverse events and impeded wound healing.”
    • This sentence seems to imply that there are frequent adverse events and that wound healing can be impeded by exercises, which is not often the case. You come back to this in line 97. Could you change this sentence?

DE – Thank you to the Reviewer for this comment. We have adjusted the sentences (line 47 and 97) to reflect the possible application of intensity monitoring and interplay with infrequent and low likelihood adverse events. The statements were included to strengthen the study rationale and support the study aim.

Line 184:

    • Which outcome parameter was used to calculate power (the mean RPE or the TL)? Was this calculated in advance?

DE – Thank you for the query. Published session RPE value dispersion values (SEE) were used to estimate the sample size. Manuscript updated to add detail and reference.  

Line 66 and in the Methodology Section:

    • Were there any specific protocols or training provided to the participants to standardize their responses to the RPE?

DE - Thank you for the query. With regards to the methodology of this study we have added details about how participants were familiarised to the RPE scale in Methods. This occurred prior to the start of the first session.

  1. Line 365-380:
    • It is indeed curious that no association was found between pain and TL in your study. Exercise can also have a pain-reducing effect; could this have contributed to your findings?

DE - This was indeed interesting. We can only speculate as to why no association was noted between pain and TL though in previous studies by our group (Gittings et al, 2020 - The efficacy of resistance training in addition to usual care for adults with acute burn injury: A randomised controlled trial), we have noticed that strength training does reduce pain for a good proportion of patients. Thus, perhaps exercise did have a pain reducing effect, also the smaller mean %TBSA may influence the statistics such that we did not observe a significant association between pain and TL in this study.

This study provides valuable insights that can improve the monitoring and prescription of exercise in burn rehabilitation, ultimately benefiting patient outcomes. 

DE – Thank you.

Reviewer 2 Report

Comments and Suggestions for Authors

Thank you for considering EBJ for the publication of your article, "Session Rating of Perceived Exertion is a Valid Method to Monitor Intensity of Exercise in Adults with Acute Burn Injuries."

General Comments:

This is an exceptionally well-written article that effectively addresses a critical topic. The study provides a thorough discussion of the utility of the current validation analysis in a clinical population and highlights the scope for further research, particularly regarding the impact of pre-exercise pain.

The research question is clearly formulated and supported by robust evidence regarding the session rating of perceived exertion. The topic is highly original and makes a significant contribution to the field by bridging a gap in exercise intensity monitoring for adults with acute burn injuries. The article offers a fresh perspective and valuable insights into this area of clinical research.

From our perspective, no improvements are necessary, and we are pleased to accept the article in its original form. The conclusions are well-founded and comprehensively address the research question. Additional tables or figures are not required to enhance the manuscript.

We have no further comments.

Author Response

Reviewer #2

This is an exceptionally well-written article that effectively addresses a critical topic. The study provides a thorough discussion of the utility of the current validation analysis in a clinical population and highlights the scope for further research, particularly regarding the impact of pre-exercise pain.

DE – Thank you to the Reviewer for these encouraging remarks.

The research question is clearly formulated and supported by robust evidence regarding the session rating of perceived exertion. The topic is highly original and makes a significant contribution to the field by bridging a gap in exercise intensity monitoring for adults with acute burn injuries. The article offers a fresh perspective and valuable insights into this area of clinical research.

DE – Again, thank you to the Reviewer for their comments.

From our perspective, no improvements are necessary, and we are pleased to accept the article in its original form. The conclusions are well-founded and comprehensively address the research question. Additional tables or figures are not required to enhance the manuscript.

We have no further comments.

DE - Thank you to the Reviewer for your time in reviewing this manuscript it is greatly appreciated.

Reviewer 3 Report

Comments and Suggestions for Authors

Thanks for the invitation.

This study explored the use of session rating of perceived exertion in the burn population.

Major concern:

1.     The title highlights ‘acute burn injuries’, however, the participant criteria only specify ‘adult’ and ‘receiving acute inpatient or post-discharge outpatient physiotherapy’. Although table 1 shows participants’ average days since burn were 28-32 (SD 63-65), a clear definition of acute burns is needed.

2.     One major aim is ‘assess if sRPE or TL have a relationship to objective measures of intensity (HR and BLa)’. But the results showed insignificant correlations, the authors then turned to using the high correlations with PRE to support its validity. Also, the methods description (line 23-26) in the abstract more likes this study is to test the validity of RPE, and in the tables ‘average RPE’, ‘session RPE’ and ‘RPE average for session’ are quite confusing. Recommend to update the aim and use sRPE (or session RPE) consistently

3.     In the analysis, ‘Independent covariates that demonstrated statistically significant association with both outcome and predictor variables were considered as confounders for inclusion in multivariable models.’ As no common independent covariates were identified, what are the reasons to include ‘age’ and TBSA’ in the models?

Author Response

Reviewer #3

Thanks for the invitation. This study explored the use of session rating of perceived exertion in the burn population.

Major concern:

  1. The title highlights ‘acute burn injuries’, however, the participant criteria only specify ‘adult’ and ‘receiving acute inpatient or post-discharge outpatient physiotherapy’. Although table 1 shows participants’ average days since burn were 28-32 (SD 63-65), a clear definition of acute burns is needed.

DE – The Reviewer makes a good point. Table 1 has been replaced to include median and IQR measures of centrality and dispersion as a number of characteristics were markedly skewed and misleading to the reader. Most patients recruited in the cohort were in fact acute burn inpatients. I believe this will clarify the situation and support the title to remain unchanged.

  1. One major aim is ‘assess if sRPE or TL have a relationship to objective measures of intensity (HR and BLa)’. But the results showed insignificant correlations, the authors then turned to using the high correlations with PRE to support its validity. Also, the methods description (line 23-26) in the abstract more likes this study is to test the validity of RPE, and in the tables ‘average RPE’, ‘session RPE’ and ‘RPE average for session’ are quite confusing.

DE – Thank you for this comment. Average RPE was the calculated mean of all the patient’s RPE ratings recorded during the gym workout (see Methods section 2.2.2) whereas the sRPE was a one-off rating of the overall session exertion by the patient and the supervising clinician. The Abstract and Methods sections have been adjusted to improve clarity.  

Recommend to update the aim and use sRPE (or session RPE) consistently.

DE – Thank you. We have defined session RPE with greater clarity in Abstract and changed “Session RPE” in Table 2 and Table 3 to “sRPE” for consistency.

  1. In the analysis, ‘Independent covariates that demonstrated statistically significant association with both outcome and predictor variables were considered as confounders for inclusion in multivariable models.’ As no common independent covariates were identified, what are the reasons to include ‘age’ and TBSA’ in the models?

DE – Thank you to the Reviewer for this query. The decision to include ‘age’ and ‘TBSA’ as covariates based on clinical reasoning in that both have been identified as impacting the metabolic responses after a burn and therefore, we ‘forced’ the adjustment in the multivariable models to improve the interpretability of the analyses.

Round 2

Reviewer 3 Report

Comments and Suggestions for Authors

It looks good.